# Positive Effects of a Perovskite Film on the Radioluminescence Properties of a ZnO:Ga Crystal Scintillator

**DOI:** 10.3390/ma15041487

**Published:** 2022-02-16

**Authors:** Shiyi He, Yang Li, Liang Chen, Tong Jin, Linyue Liu, Jinlu Ruan, Xiaoping Ouyang

**Affiliations:** 1State Key Laboratory of Intense Pulsed Radiation Simulation and Effect, Northwest Institute of Nuclear Technology, Xi’an 710024, China; heshiyi@nint.ac.cn (S.H.); liulinyue@nint.ac.cn (L.L.); ruanjinlu@nint.ac.cn (J.R.); ouyangxiaoping@nint.ac.cn (X.O.); 2The Department of Nuclear Science and Engineering, Nanjing University of Aeronautics and Astronautics, Nanjing 210016, China; 18811617658@163.com; 3Wuhan National Laboratory for Optoelectronics and School of Optical and Electronic Information, Huazhong University of Science and Technology, Wuhan 430074, China; tongjin@hust.edu.cn

**Keywords:** ZnO:Ga (GZO) crystal scintillator, perovskite film, radioluminescence (RL) properties, two-layer structure, imaging

## Abstract

To improve the radioluminescence (RL) performance of ZnO:Ga (GZO) crystal scintillators and overcome the challenge of their self-absorption, we proposed a two-layer composite scintillator consisting of a GZO wafer and a 70 nm lead halide perovskite film(CsPbBr_3_, CH_3_NH_3_PbBr_3_). The effects of the perovskite film on the RL properties were studied. The results showed that the perovskite quantum dot film substantially changed the RL spectrum of GZO and prevented self-absorption. The RL of the samples were enhanced by 66% to 151% through the photoluminescence (PL) of the perovskite film, while the energy-resolving power and spatial-resolving power were maintained at the same level as that of GZO image converters. The present experiments and discussions confirmed that the perovskite film improved the RL, and this study suggests a new wavelength regulation method among scintillators, converters, and back-end optical devices. The applications of perovskites in the field of radiation detection and imaging have been extended.

## 1. Introduction

Organic–inorganic halide perovskites, a class of materials with great potential in both the semiconductor and scintillator communities [1,2], have been applied in various scientific research fields, such as optical communication, solar cells, sensing, radiation detection, and imaging [3,4,5,6].

As scintillator materials, perovskites effectively absorb light in a wide range from the ultraviolet to the visible and near-infrared region [7] and yield light of various wavelengths. In previous works [8,9], the absorption efficiency of lead halide perovskites for 250 nm to 450 nm light was over 97%. The wavelengths of the output light were 90% concentrated in the range from 500 nm to 570 nm. Large absorption coefficients indicated that a hundred-nanometer layer of perovskite was sufficient for almost complete light absorption.

When excited by an X-ray beam, perovskites can also yield color-tunable emissions. Thus, perovskites, especially perovskite quantum dots (QDs), have received much attention for their X-ray scintillation properties [10,11,12,13]. For instance, CsPbBr_3_/Cs_4_PbBr_6_ NCs exhibited desirable scintillation properties with a 10 ns decay time, a 3.0 ± 0.1% energy resolution, and a 64,000 photons/MeV light yield [14]. Their application value is rising swiftly in X-ray detection and imaging [15,16].

However, many perovskites have a higher response to UV light [17]. Organic manganese (II) halide hybrids have been reported to exhibit strong photoluminescence (PL) with colors ranging from green to red [18,19]. Comparatively, these perovskites are more suitable for UV-light detection. The applications of these perovskites in X-ray detection and imaging may need to be considered in composite ways [20,21,22]. As reported, a new type of liquid scintillator by hybridizing CsPbX_3_ (X: Cl, Br, I) nanocrystals was reported with a high quantum yield and a 3.5 Lp/mm image resolution.

As a mature scintillator material, ZnO has a sub-nanosecond ultrafast response [23] and was successfully used in radiation detection [24,25]. Additionally, high temporal resolution-imaging results were obtained [26,27]. However, ZnO:Ga (GZO) and many other ultrafast scintillator materials have two common shortcomings: high self-absorption and an untunable, narrow radioluminescence (RL) spectrum [28,29]. Thus, the scintillation efficiency is low and hard to promote by increasing the thickness, which restrains the detection and imaging quality. In addition, the external quantum efficiency of GZO is low. Specifically, the index of refraction of GZO is greater than 2.0.

Many methods of increasing the scintillation efficiency have been applied [30,31,32], such as nanoimprint, annealing, and photonic crystal. Along these lines, a photonic crystal structure was used to cover the incident surface of a GZO crystal [33]. The light yield of the crystal was increased by 59% to 84%. However, fabrication of microstructures on GZO surfaces is very difficult. The photonic crystal structure layer brought more structural defects and a complicated modulation transfer function (MTF). In addition, the photonic crystal structure did not change the wavelength of the light, and the reflected light was still absorbed by the GZO crystal.

An idea was proposed to coat a thin perovskite film instead of a photonic crystal structure on a GZO scintillator. This approach is expected to improve the luminous quality. GZO crystals output light with wavelengths of 380 nm to 400 nm, which are in the absorption wavelength range of perovskites. CsPbBr_3_, CH_3_NH_3_PbBr_3_ (MAPbBr_3_), or other perovskites output light with wavelengths of 450–600 nm, which are outside of the absorption wavelength range of GZO. Additionally, surface treatment of GZO crystals with a perovskite film is more flexible, and optimization of the external quantum efficiency or MTF curve is easier.

In this Letter, lead halide perovskites were spin-coated on the surface of a GZO crystal scintillator. Comparisons were made between pure GZO crystals and those coated by perovskite films in terms of the RL properties. RL spectra excited by X-rays and lifetime curves and photon energy responses excited by alpha particles were measured to acquire the RL properties of composite scintillators in different layouts. The basic effects of the perovskite film on the spatial-resolving power, uniformity, and MTF curves were studied. Furthermore, a data summary and a discussion focusing on the RL yield were provided to explain the results theoretically and ensure the consistency with former experiments.

## 2. Materials and Methods

GZO single crystals were fabricated by the State Key Laboratory of Optoelectronic Materials and Technologies, School of Materials, Sun Yat-sen University. A hydrothermal method was used to grow a high-quality gallium-doped ZnO bulk single crystal [34]. Two-layer composite scintillators were successfully produced through high-temperature mixing, suspension coating, and negative pressure drying, as shown in Figure 1a. For more details, CsPbBr_3_ and MAPbBr_3_ film were fabricated by modified precipitation methods [35,36], respectively.

Scanning electron microscopy (SEM) was carried out with an Oxford (X-ACT, Oxford, Oxon, UK) scanning electron microscope. X-ray diffraction (XRD) patterns were measured through a diffractometer (Ultima IV, Rigaku, Tokyo, Japan) equipped with a Cu Kα X-ray tube (λ = 0.154 nm). Transmission spectra were measured with a UV spectrophotometer (Lamda-900, PerkinElmer, Waltham, MA, USA).

The RL spectra of GZO and composite samples were measured. An X-ray generator (12 W X-ray source, Moxtek, Orem, UT, USA), a photomultiplier tube (PMT, 77360, Newport, Irvine, CA, USA), a spectrometer (74126, Newport, Irvine, CA, USA), and a power meter (1936-R, Newport, Irvine, CA, USA) were used. The spectrometer was connected to a PMT. The spectral response of the PMT is shown in Figure 1a.

A diagram showing the homemade time-correlated single-photon-counting (TCSPC) system was illustrated, as shown in Figure 1b. The system mainly contained a microchannel plate (MCP, R3809U-52, Hamamatsu, Hamamatsu City, Japan) for single proton signal, a PMT (9815, ET Enterprises Limited, London, UK) for signal acquisition, and a series of electronic devices, including a time-to-amplitude converter (TAC, 567, ORTEC, Oak Ridge, TN, USA), a constant-fraction discriminator (935, ORTEC, Oak Ridge, TN, USA), two preamplifiers (9306, ORTEC, Oak Ridge, TN, USA), a pico-timing discriminator (9307, ORTEC, Oak Ridge, TN, USA), a delay unit (425, ORTEC, Oak Ridge, TN, USA), and a computer-controlled multi-channel analyzer (MCA8000A, Amptek, Bedford, MA, USA). A neptunium (Np) source producing 4.7 MeV alpha particles was used to excite the samples. High voltages were supplied to the PMTs by two high-voltage DC power systems (PS350, Stanford Research Systems, Sunnyvale, CA, USA).

The energy resolutions and output luminescence intensities of the samples were measured by the pulse height spectrum system. The system contained a PMT (CR173-Q1, Hamamatsu, Hamamatsu City, Japan), a scintillation preamplifier (113, ORTEC, Oak Ridge, TN, USA), a spectroscopy amplifier (672, ORTEC, Oak Ridge, TN, USA), and a multi-channel (MCA8000A, Amptek, Bedford, MA, USA). The spectral response of the PMT is shown in Figure 1c.

The X-ray-imaging system consists of an X-ray generator (12 W X-ray source, Moxtek, Orem, UT, USA), a mirror, and a professional camera (DH34-18F-63, Andor, Belfast, UK). The operated voltage was 30 kV. The dose rates were altered by the input current. A photograph of a standard pattern plate is shown in Figure 1d. The resolution of the line pairs in the red box was 1 Lp/mm.

## 3. Results

GZO single crystal wafers, as shown in Figure 2a, were prepared by the hydrothermal method [37]. The thickness of the wafers was 0.40 ± 0.03 mm, and the carrier concentration was ~1 × 10^19^ cm^−3^ [38,39]. According to the intuitive ideas shown in Figure 2b, two-layer composite scintillators were successfully produced.

A real picture of samples is shown in Figure 2c. One thinner perovskite foil was coated on GZO single crystal wafers. The micrograph of the two-layer sample in Figure 2d shows that the thickness of the perovskite foil was ~65 nm and that the two layers fit closely.

The X-ray diffraction (XRD) pattern proved that MAPbBr_3_ QDs truly existed in the thin foil. CsPbBr_3_, or other QDs, could also serve as the luminescence center.

The working principle of the composite samples is shown in Figure 3a. X-rays simultaneously act on GZO and the perovskite film. Both are body sources for luminescence. As the light path shows in the 01# area, the output light contains three components on the perovskite side: (a) the light produced by the perovskite film, (b) the light produced by GZO and transformed by the perovskite film, and (c) the light produced by GZO but that failed to be transformed by the perovskite film. As shown in the 02# area, the output light also contains three components on the GZO side: (a) the light produced by the perovskite film and transported through GZO, (b) the light produced by GZO and emitted forward, and (c) the light produced by GZO but emitted backwards to the perovskite film, then transformed by the perovskite film, and finally emitted forward.

The RL spectra of separate GZO and PL spectra of MAPbBr_3_ are shown in Figure 3b, as well as the PL spectra of CsPbBr_3_. Their PL peaks were located at 526 nm and 513 nm. Transmission spectrum tests were carried out and the results are shown in Figure 3c. GZO uniformly transmitted the perovskite PL spectrum, while the RL spectrum of GZO was uniformly absorbed by the perovskite film.

Further experiments on the RL spectrum excited by X-rays were conducted, and the results are shown in Figure 3d. Four conditions were considered, including both sides of the composite scintillator, only GZO, and only the perovskite film. The spectral response of converters was deducted.

After the perovskite film was coated, a luminescence peak of the perovskite film appeared in the spectra of both sample sides. The ratio of the peak in the RL spectrum of GZO was suppressed. The wavelength conversion was significant, especially on the perovskite film side.

The output light spectra of both sides of the samples were fitted by spectra of only GZO and only perovskite film. On the GZO side, the area ratio was 0.63. On the perovskite side, the area ratio was 0.036. Here, the PL of the perovskite was the main component and that of GZO was almost all absorbed and transformed.

The decay times of the composite samples are shown in Figure 4a. A single photon acquisition system was set on the GZO side since the perovskite film absorbed almost all the GZO RL. Different from X-ray excitation conditions, the RL sample excited by alpha particles was a point source and could statistically be treated as one area source near the surface. Correspondingly, the decay of each kind of luminescence in GZO, especially its own luminescence, should be considered. As shown in Figure 4b, the test should be divided into four cases according to the directions from which alpha particles arrive and in which photons are collected.

For convenience, the following abbreviations are used. When alpha particles were incident from the perovskite side and were detected on the GZO side, these samples were abbreviated as PiZo. The remainder of the samples were abbreviated as PiPo, ZiPo, and ZiZo.

The decay time of the GZO wafer was 1.2 ns. With the perovskite film coating, a slow component was observed in the decay time curve of the samples, that is, 13.7 ns from MAPbBr_3_ and 9.9 ns and 37.2 ns from CsPbBr_3_.

Moreover, the intensity ratio of the GZO RL to the perovskite, PL, was deduced from the fitting result. In the PiZo cases, the ratio was 0.35 for the CsPbBr_3_-coated samples and 0.36 for the MAPbBr_3_-coated samples. In the ZiZo cases, the ratio was 1.26 for the MAPbBr_3_-coated samples.

Comparatively, the curve marked by ‘△’ had the same components as the curve marked by ‘▽’, but the ratio was 3.3 times lower. This result was caused by the larger decay efficiency of the GZO wafer on its own RL.

Experiments on the energy resolution of the samples were conducted, as shown in Figure 5. The energy resolution of the GZO wafer was 18.4%. In the PiPo cases, the energy resolution was 15.1% for the CsPbBr_3_-coated samples and 20.3% for the MAPbBr_3_-coated samples. The energy resolution was 17.7% for the MAPbBr_3_-coated samples in the PiZo cases and 19.2% for the CsPbBr_3_-coated samples in the ZiZo cases. The energy resolutions of each case were similar, and there was no regular difference.

By deducting the spectral response curve of the photomultiplier tube (PMT), the relative intensity of the RL sample excited by α particles was obtained. Based on the RL from GZO wafers, the energy spectra of each sample are shown by the filled areas in Figure 5.

Compared with those of the peak channel of GZO, the output luminescence intensities of the composite samples were all notably increased. In the PiPo cases, the ratio of output luminescence intensity to that of GZO was 174% for the CsPbBr_3_-coated samples and 160% for the MAPbBr_3_-coated samples. The ratio was 201% for the MAPbBr_3_-coated samples in the PiZo cases and 175% for the CsPbBr_3_-coated samples in the ZiZo cases.

As an application, a series of imaging tests were performed on the samples, as shown in Figure 6. The image resolution decreased after coating but was still over 1 lp/mm. The uniformity of the luminance was 92.5%, and the half-width of the pixel position was 0.65 mm for the GZO wafer. When GZO was the image surface of the composite samples, the uniformity of the luminance and the half-width of the pixel position were 90.7% and 0.77 mm, respectively. When the perovskite film was the image surface, they were 90.3% and 0.77 mm, respectively. Values of the above structures were 2.41, 2.25, and 2.16, respectively. With the perovskite film, the imaging quality of the samples decreased slightly, but the response to X-rays was strengthened.

## 4. Discussion

Further quantitative analysis could be performed on the enhancement of GZO RL by PL of the perovskite film.

As shown in Figure 7a, the conversion process was based on the conservation of energy. Since the wavelength of the perovskite PL was longer, photon multiplication occurred in the conversion process. The photon multiplication ratio is denoted by k, and k is described in Equation (1). In this equation, $\bar{E}$ is the average luminescence energy, f(λ) is the luminescence spectrum, and $E_{i}$, $\lambda_{i}$ and $\Delta \lambda_{i}$ are the average energy, average wavelength, and interval length of the *i*th bin, respectively.

The ratios of MAPbBr_3_ and CsPbBr_3_ were close, and thus, k was set to 1.31 in later discussions. However, not every photon would be converted; thus, a conversion efficiency, α, should be denoted as well.
(1)$$k=\frac{{\bar{E}}_{Per}}{{\bar{E}}_{ZnO}}=\frac{\sum_{i=1}^{n}f_{Per}\left(\lambda_{i}\right)E_{i}\Delta \lambda_{i}}{\sum_{i=1}^{n}f_{ZnO}\left(\lambda_{i}\right)E_{i}\Delta \lambda_{i}}=\frac{\sum_{i=1}^{n}f_{Per}\left(\lambda_{i}\right)\frac{1}{\lambda_{i}}\Delta \lambda_{i}}{\sum_{i=1}^{n}f_{ZnO}\left(\lambda_{i}\right)\frac{1}{\lambda_{i}}\Delta \lambda_{i}}$$

To describe the whole process, light transmission was refined into two parts: at interfaces and inside materials. The assumption was made that transmission of light of all wavelengths was equal for one interface, but differences in the GZO absorbability of luminescence at different wavelengths were still retained. As shown in Figure 6b, for half-space uniformly distributed light, the ratios of luminescence passing through the GZO–air, perovskite–air, GZO–perovskite, and perovskite–GZO interfaces were set as $g_{A}$, $g_{D}$, $g_{1}$, and $g_{2}$, respectively. The decay ratios of light passing through the GZO wafers were set as $\eta_{D}$ for perovskite luminescence and $\eta_{A}$ for GZO luminescence. For convenience, $m_{1}=g_{1}g_{D}/2g_{A}$ and $m_{2}=g_{1}g_{2}\eta_{D}/2$ were set.

When excited by X-rays, the samples were body sources. Let the number of photons produced by the perovskite and GZO under X-ray excitation be 2Y and 2X, respectively. Thus, the single-side RL intensities of the perovskite and GZO were $D=Yg_{D}$ and $A=Xg_{A}$, respectively.

When taking the perovskite film as the image surface along with the principle of luminescence in Figure 3a, the intensity of the composite sample was $D+A\alpha km_{1}+2A\left(1-\alpha \right)m_{1}$. Here, $2A\left(1-\alpha \right)m_{1}$ was from the GZO RL spectrum, and $D+A\alpha km_{1}$ was from the perovskite PL spectrum. According to the experiments, their ratio is as described in Equation (2):(2)$$\frac{2m_{1}\left(1-\alpha \right)}{\alpha km_{1}+D/A}=0.036$$

When GZO was used as the image surface, the intensity of the composite sample was $A+Dm_{3}+A\alpha km_{2}$. According to the experiments, their ratio is as described in Equation (3).
(3)$$\frac{1}{\left(\alpha km_{2}+\frac{m_{2}}{m_{1}}D/A\right)}=0.63$$

The luminescence produced by GZO when excited by alpha particles is 2Z. Thus, the same-side PL intensity of GZO was $S=Zg_{A}$. Little energy was deposited in the perovskite film since its thickness was less than 100 nm. Combined with the principle of luminescence in Figure 4b, four cases were considered.

In PiZo cases, the luminescence contained two parts. One was the forward RL generated by the incident surface and then decayed by GZO itself. The other was the PL converted by the perovskite and then forward emitted. Thus, the intensity of the emission luminescence was $\eta_{A}S+\alpha km_{2}S$. According to the experiment results, their ratios are as described in Equation (4).
(4)$$\{\begin{array}{ll} \frac{\eta_{A}}{\alpha km_{2}}=0.35 & 2-CiZo\\ \eta_{A}+\alpha km_{2}=2.01 & 3-MiZo\\ \frac{\eta_{A}}{\alpha km_{2}}=0.36 & 2-MiZo \end{array}$$

Similarly, the emission intensity in other cases could be obtained. In PiPo cases, the intensity of the emission luminescence was $\alpha km_{1}S+2\left(1-\alpha \right)m_{1}S$. As the experiments show, their intensities are described in Equation (5).
(5)$$\{\begin{array}{ll} 2\left(1-\alpha \right)m_{1}+\alpha km_{1}=1.60 & 3-MiMo\\ 2\left(1-\alpha \right)m_{1}+\alpha km_{1}=1.74 & 3-CiCo \end{array}$$

In the ZiZo case, the intensity of the emission luminescence was $S+\eta_{A}\alpha km_{2}S$, and the luminescence ratio of GZO and the perovskite was $S/\eta_{A}\alpha km_{2}S$. Their intensities and ratios are described in Equation (6).
(6)$$\{\begin{array}{ll} 1+\eta_{A}\alpha km_{2}=1.75 & 3-ZiZo\\ 1/\left(\eta_{A}\alpha km_{2}\right)=1.26 & 2-ZiZo \end{array}$$

To date, nine equations and five unknowns have been acquired. The Gauss-Newton method was used to optimize the solution of the nonlinear overdetermined equations. Two sets of approximate solutions were obtained, as shown in Equation (7), by fixing D/A=0 or not.
(7)$$\left(\alpha ,\eta ,m_{1},m_{2},D/A\right)=\{\begin{array}{ll} \left(0.98,1.26,1.18,0.53,0.00\right) & D/A=0\\ \left(0.98,1.26,1.16,0.53,0.11\right) & D/A>0 \end{array}$$

Except for D/A, there was little change in the parameters of the two cases. Therefore, ignoring the perovskite film RL in the experiment is appropriate, as are the hypotheses. Taking the parameters back into the equations, the average error was only 2.36%.

As a result, under X-ray excitation, when taking the perovskite film as the image surface, the luminous intensity of the composite sample was 1.66 times that of pure GZO. When taking the GZO film as the image surface, the luminous intensity of the composite sample was 2.51 times that of pure GZO.

The luminous intensity of the composite sample was 2.04 times that of pure GZO in the PiZo cases, 1.66 times that of pure GZO in the PiPo cases, 1.80 times that of pure GZO in the ZiZo cases, and 0.88 times that of pure GZO in the ZiPo cases.

## 5. Conclusions

To improve the RL performance of GZO crystal scintillators and overcome their self-absorption problem, a piece of lead halide perovskite (CsPbBr_3_, MAPbBr_3_) film was spin-coated on the surface. This way, a kind of two-layer composite scintillator was successfully made and primarily studied. The enhancement on RL brought by perovskite film was higher than the reported results of nanoimprint (30%) and photonic crystal (59~84%).

The RL spectra on both sides of the two-layer composite scintillator were changed. Especially on the perovskite side, a total of 98% of the GZO luminescence was converted to perovskite luminescence. The light yield of the ZnO crystal scintillator was notably enhanced after a piece of the perovskite film was coated. Whether excited by X-rays or alpha particles, the RL intensity increased by 1.6 to 2.5 compared with only GZO. Compared with GZO alone, the primary factor limiting the decay time of the crystals was the decay time of the perovskite QDs. CsPbBr_3_ performed better since its main component in the decay time curve was less than 10 ns. No explicit difference in energy resolution from only GZO samples was noted. Composite samples could serve as an image convertor on both sides. The uniformity of the composite samples was 90%, and MTF0.1 = 2.2, which were at the same level as those of GZO alone (92%, 2.4).

## Figures and Tables

**Figure 1 materials-15-01487-f001:**
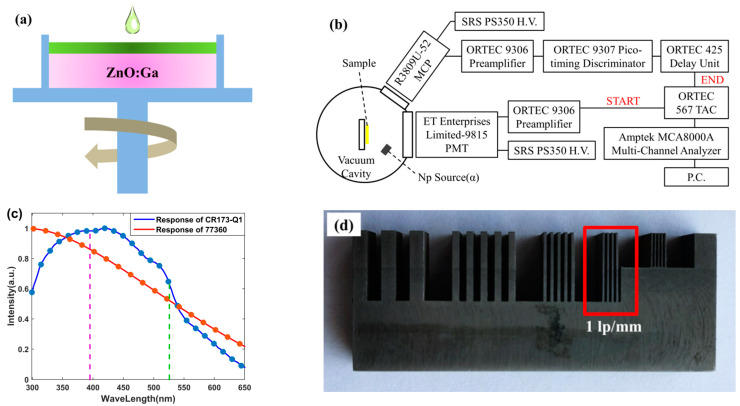
(**a**) Schematic diagram of the composite preparation. (**b**) Diagram of the TCSPC system. (**c**) Spectral response of the Newport-77360 PMT and the Hamamatsu CR173-Q1 PMT. (**d**) Photograph of a standard pattern plate.

**Figure 2 materials-15-01487-f002:**
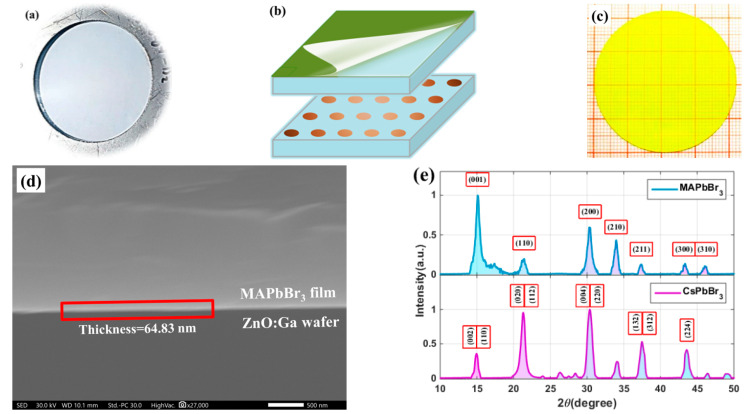
Pictures and structures of two-layer composite samples: (**a**) ZnO:Ga (GZO) wafer; (**b**) two typical combination ideas; (**c**) two-layer composite sample; (**d**) micrograph of the two-layer sample; (**e**) XRD pattern of the perovskite film and GZO.

**Figure 3 materials-15-01487-f003:**
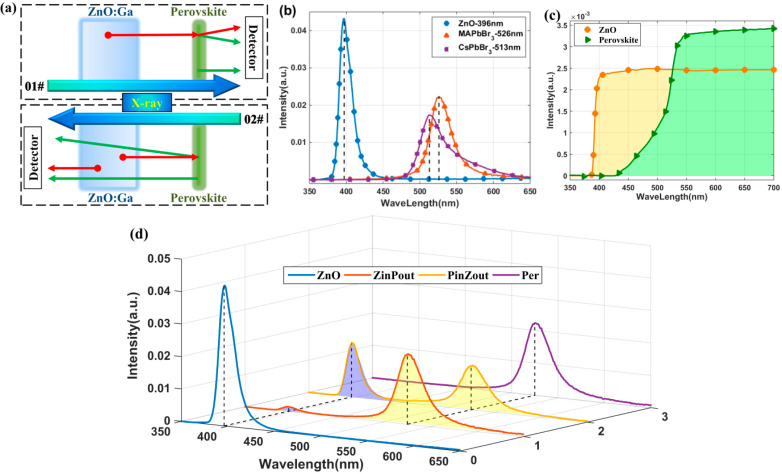
Luminescence properties of GZO, the perovskite, and the two-layer composite scintillator: (**a**) working principles of two-layer composite samples; (**b**) RL spectra of GZO and PL spectra of perovskite films; (**c**) transmission spectra of samples; (**d**) RL curves of composite samples and their fitting.

**Figure 4 materials-15-01487-f004:**
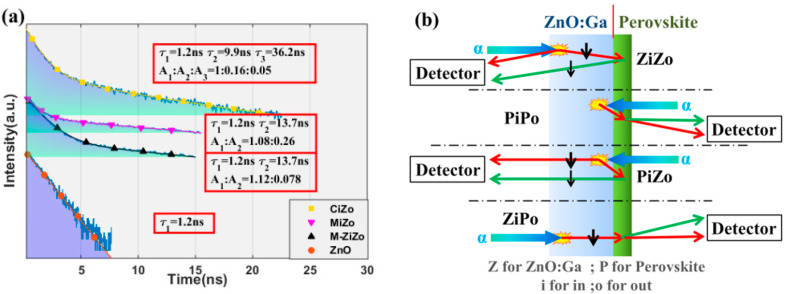
Decay times measurements of excitation by alpha particles collected from the GZO side: (**a**) decay curves of different samples and layouts; (**b**) optical transmission diagrams for 4 conditions.

**Figure 5 materials-15-01487-f005:**
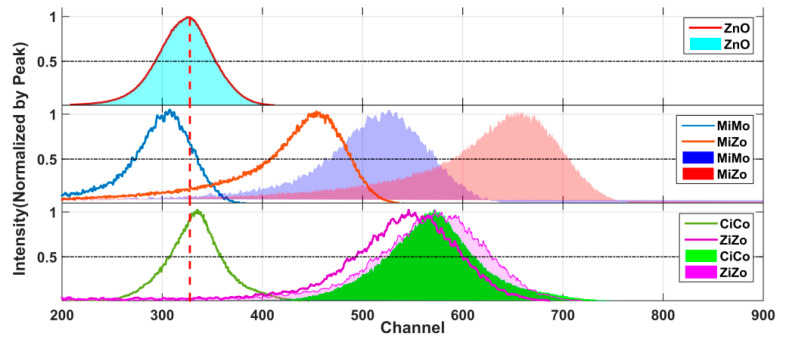
Energy spectra of various samples excited by α particles before and after deducing the spectral response.

**Figure 6 materials-15-01487-f006:**
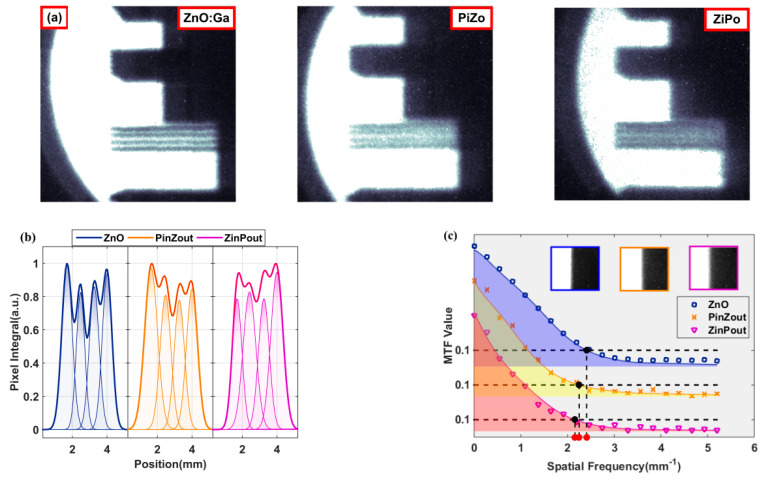
Imaging properties of samples: (**a**) greyscale photos of the resolution card for GZO wafers, composite samples taking GZO as the image surface, and composite samples taking the perovskite film as the image surface; (**b**) integral of the same part of the greyscale photos and pixel resolution fitting; (**c**) MTF curves from knife edge pictures.

**Figure 7 materials-15-01487-f007:**
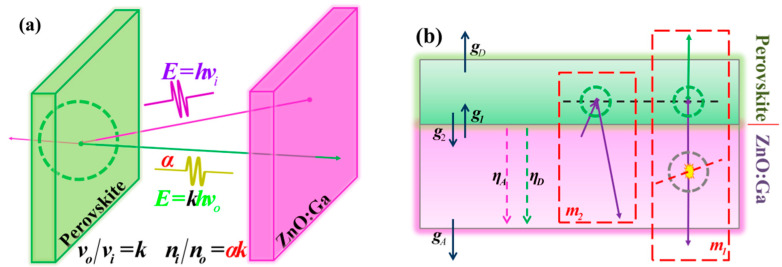
Schematic diagram for theoretical analysis: (**a**) conversion process of photons in the perovskite film; (**b**) quantitative optical path analysis.

## Data Availability

The data supporting the reported results are not stored in any publicly archived datasets. Readers can contact the corresponding author for any further clarification of the results obtained.

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
