# Peer review of "Positive Effects of a Perovskite Film on the Radioluminescence Properties of a ZnO:Ga Crystal Scintillator"

_materials, 2022, doi:10.3390/ma15041487_

Round 1

Reviewer 1 Report

I found the work certainly interesting and found all the discussion and conclusions sound and sensible. The manuscript was also well-written and properly organized.

Certain words are highlighted and comments have been given. Please make those changes and then submit the manuscript. The pdf file is attached.

Author Response

Point 1: Certain words are highlighted and comments have been given. Please make those changes and then submit the manuscript.

Response 1: We appreciate your affirmation and professional review. Thank you very much for pointing out those errors. We have modified all of six errors and double-checked the rest of the manuscript.

Reviewer 2 Report

The authors have prepared ZnO:Ga (GZO) crystal scintillators to improve radioluminescence (RL) performances and overcome their self-absorption problems. However, the following comments need to be addressed before considering publication in materials-1578726.  

Comments:           

  • More explanations, recent achievements, and corresponding citations are necessary for the introduction section.
  • For better understanding for readers, authors should provide a schematic diagram of the composite preparation.
  • Moreover, the authors provide an explanation with recent compared reported results.  

Author Response

Point 1: More explanations, recent achievements, and corresponding citations are necessary for the introduction section.

Response 1: Thank you very much for your constructive review. We have added more explanations and achievements in the aspects of X-ray scintillation properties of perovskite, UV light detection properties and its cross applications with other materials. Corresponding references were added in the manuscript [1-4]. Furthermore, changes will be made from this aspect in the subsequent writing of other articles, especially the introduction.

  1. Morad, V.; Shynkarenko, Y.; Yakunin, S.; Brumberg, A.; Schaller, R. D.; Kovalenko, M. V. Disphenoidal zero-dimensional lead, tin, and germanium halides: Highly emissive singlet and triplet self-trapped excitons and X-ray scintillation. Am. Chem. Soc. 2019, 141, 9764–9768, https://doi.org/10.1021/jacs.9b02365
  2. Xu, Q.; Wang, J.; Shao, W.; Ouyang, X.; Wang, X.; Zhang, X.; Guo, Y.; Ouyang, X. A solution-processed zero-dimensional all-inorganic perovskite scintillator for high resolution gamma-ray spectroscopy detection. Nanoscale 2020, 12, 9272–9732, https://doi.org/10.1039/ D0NR00772B
  3. Li, M.; Zhou, J.; Molokeev, M. S.; Jiang, X.; Lin, Z.; Zhao, J.; Xia, Z. Lead-free hybrid metal halides with a green-emissive [MnBr4] unit as a selective turn-on fluorescent sensor for acetone. Chem. 2019, 58, 13464–13470, https://doi.org/10.1021/acs.chemmater.9b03782
  4. Morad, V.; Chenriukh, I.; Pőttschacher, L.; Shynkarenko, Y.; Yakunin, S.; Kovalenko, M. V. Manganese(II) in tetrahedral halide environment: factors governing bright green luminescence. Mater. 2019, 31, 10161–10169, https://doi.org/10.1021/acs.chemmater.9b03782

Point 2: For better understanding for readers, authors should provide a schematic diagram of the composite preparation.

Response 2: Thank you very much for your professional suggestion. We have re-orgnized the Figure 1. A schematic diagram of the composite preparation was illustrated in Figure 1a. Two Response curves of photoelectric multiplier tube were combined into one, as shown in Figure 1c.

Point 3: Moreover, the authors provide an explanation with recent compared reported results.

Response 3: Thank you very much for your suggestion. Recent compared reported results were arranged in introduction. And the corresponding statements have been supplemented in conclusion as follows: “The enhancement on RL brought by perovskite film was higher than the reported results of nanoimprint (30%) and photonic crystal (59%~84%).”

Reviewer 3 Report

Manuscript #: materials-1578726

Title: Positive Effects of a Perovskite Film on the Radioluminescence of Properties of a ZnO:Ga Crystal Scintillator

Comments:

The manuscript describes the radioluminescence characteristics of ZnO:Ga crystal scintillator with a bilayer structure with perovskite films. The authors provide experimental results and analysis to claim the advantage of the perovskite coating.   The manuscript is well-written and informative. Below is the list of a few comments.

  1. Can the authors give some more explanation in Fig. 1(b)?
  2. Please, add film names on the SEM image in Fig. 1(d).
  3. Can the authors add # of the light path in Fig. 3(a) with a modification of line 127 – 136?
  4. Please, give more explanation why the PiZo and ZiPo image quality is lower compared to the ZnO:Ga one in Fig. 6(a)?

Author Response

Point 1: Can the authors give some more explanation in Fig. 1(b)

Response 1: Thank you very much for your patient and professional review. For each excitation in experiment, PMT collected most of the scintillation photons and its output was discriminated to provide the “start” timing signal. The MCP worked on single photon counting mode and its signal was amplified, discriminated and delayed to provide the “stop” timing signal. The time delay between two timing signals was recorded by TAC and registered by multi-channel analyzer. Thus, the decay time curve of scintillator was obtained. As a mature method, single photon method has a wide range of applications. Iit was described in more detail in reference 1.

  1. Liu, J.; Liu, F.; Ouyang X.; Liu, B.; Chen, L.; Ruan, J.; Zhang, Z.; Liu, J. The luminescence characteristics of CsI(Na) crystal under α and X/γ excitation. J. Appl. Phys. 2013, 113, 023101, https://doi.org/10.1063/1.4773528

Point 2: Please, add film names on the SEM image in Fig. 1(d).

Response 2: Thank you very much for your suggestion. Figure 1d was re-plotted and two film names were added.

Point 3: Can the authors add # of the light path in Fig. 3(a) with a modification of line 127 – 136?

Response 3: Thank you very much for your suggestion. Figure 3a was re-plotted and two dashed boxes and numbers of the light path were added. Modifications were conducted in the description of the working principle of the composite samples in new manuscript.

Point 4: Please, give more explanation why the PiZo and ZiPo image quality is lower compared to the ZnO:Ga one in Fig. 6(a)?

Response 4: Thank you very much for your professional review. The luminescence of perovskite film came from the inner QDs, which was a process of absorption and secondary PL. In my opinion, the inhomogeneity of the QDs distribution and the distribution of secondary PL (uniform angular distribution) produced scattering effects. Meanwhile, perovskite films were prepared by suspension coating method. In this way, the image surface was unpredictable in the ZiPo cases and resulted in a decrease of imaging quality. In the PiZo cases, the optical path did not pass through the outer surface of perovskite. As a result, the imaging quality in the PiZo cases were better than that of ZiPo cases.